# A Scalable Program for Improving Physical Activity in Older People with Dementia Including Culturally and Linguistically Diverse (CALD) Groups Who Receive Home Support: A Feasibility Study

**DOI:** 10.3390/ijerph20043662

**Published:** 2023-02-18

**Authors:** Den-Ching A. Lee, Terry P. Haines, Michele L. Callisaya, Keith D. Hill

**Affiliations:** 1Rehabilitation Ageing and Independent Living (RAIL) Research Centre, Monash University, Frankston 3199, Australia; 2National Centre for Healthy Ageing (NCHA), Monash University and Peninsula Health, Frankston 3199, Australia; 3School of Primary and Allied Health Care, Faculty of Medicine, Nursing and Health Sciences, Monash University, Frankston 3199, Australia; 4Peninsula Clinical School, Monash University, Frankston 3199, Australia

**Keywords:** cognitive impairment, culturally and linguistically diverse, CALD, dementia, exercise, falls, home care, physical activity, physical function, care support workers

## Abstract

Home care clients with dementia/cognitive impairment are typically functionally dependent and physically inactive. We pilot-tested a co-designed physical exercise program for its feasibility, safety, adherence and potential for benefits on physical activity, physical function, healthcare use and falls. Trained community care support workers delivered a 12-week home exercise program to clients with dementia/cognitive impairment, once weekly for 15 min during care shifts, supplemented by carers’ supervision of exercises for 30 min, three times weekly. A physiotherapist provided fortnightly phone support to ensure safety and exercise progression. Baseline and Week 12 assessments using validated scales for physical activity, physical function, daily living independence, falls efficacy, quality of life, self-reported healthcare use, falls and sleep quality were undertaken. Differences were examined with regression analyses. Care support workers (*n* = 26) and client/carer dyads (*n* = 26 and 80.8% culturally and linguistically diverse) participated. Participants recorded adverse events/falls and exercises in dairies. Fifteen dyads completed the program. No falls/adverse events occurred with the exercises. The adherence rates against targets for exercise time completed and days in which exercise were undertaken for support workers were 137%/79.6%, and for client/carer dyads were 82%/104.8%, respectively. Physical activity participation, physical function and falls efficacy significantly improved at Week 12 compared to baseline. The feasibility, safety and adherence of the co-designed physical exercise program were demonstrated. Strategies to minimise dropouts in future effectiveness studies are required.

## 1. Introduction

There are around one million older Australians receiving home care nationally each year [1]. The Australian aged care system provides funding for home care which enables older people who are experiencing functional and/or mental health difficulties to receive assistance to support their living at home [2], including those with cognitive impairment. The services providing through home care include support services such as domestic assistance and personal care. Care support workers who have received vocational training deliver these services. People who rely on home care represent a subgroup of older people who are often frail and functionally dependent, and therefore are at an increased risk of physical inactivity and falls [3].

Previous research identified that this population had a lower level of physical activity compared to similar older adults who did not use home care [4]. Furthermore, older people who used home care fell 50% more frequently than their counterparts who did not [5]. In Australia, over 90,000 people receiving home care were also eligible for the dementia and cognition home care supplement [2], which is an additional fund provided by the Government that helps home care providers with the cost of caring for people who have moderate to severe cognitive impairment [6]. This population is at a particularly high risk of physical inactivity and falls that have a negative impact on independence, function, mobility and quality of life [7]. 

Physical activity is beneficial for older people with frailty or dementia [8,9]. A systematic review of 13 randomised controlled trials found that physical activity interventions improved physical functions such as walking, mobility and balance and may improve quality of life in older people with dementia across hospital, residential care and community settings [10]. There is emerging evidence in another review that exercises may also reduce falls in older people with dementia living in the community [11]. Physical activity improves physical function, which may reduce the modifiable factors associated with an increased likelihood of future falls in these people, such as balance problems, reduced lower limb strength, functional dependence, gait and mobility issues. Various forms of physical activity, for example, multimodal exercise training to improve muscle strength, balance and fitness, or single modality exercise training in one of these areas, can be implemented safely at home or in the community, and achieve good outcomes in people with dementia. However, as previous research works have demonstrated, there are very low levels of physical activity participation in people who receive home support [5] and people with dementia [12]. 

Home care clients frequently receive visits from care support workers who have the potential to facilitate participation in physical activity. Another benefit of utilising care support workers to promote physical activity is that they are familiar with their clients and family carers. However, there is no minimum requirement for these workers to be trained in how to promote physical activity in people with dementia. For this to happen in the context of supporting people with dementia, it would require care support workers to be adequately equipped with skills, confidence and the capability to help people with dementia engage in physical activity, and the physical activity being promoted to be safe, effective and acceptable to people with dementia. 

For this purpose, we co-designed a physical exercise program, entitled “*Safe Functional Home Exercise” program for people with dementia* (see Appendix A) with an expert advisory panel comprising exercise and fall prevention researchers, physiotherapists, aged care staff and consumer stakeholders. This was a 12-week exercise program that used a simplified approach with a set of four exercises with no complex decision-making criteria on exercise selection and exercise progression required, designed for care support workers to implement safely. 

In this study, we aimed to pilot-test this co-designed exercise program delivered by trained care support workers for feasibility, safety, adherence and potential for benefits on physical activity, physical function, healthcare use and falls.

## 2. Materials and Methods

### 2.1. Design

A one-group pre-test–post-test design was used. 

### 2.2. Participants

There were two groups of participants: Group A comprised care support workers who were willing to: (i) attend a training course for the co-designed exercise program; (ii) supervise home care clients with dementia/cognitive impairment to exercise once weekly during their regular care shifts; and (iii)participating community aged care provider organisations supported the occurrence of this activity during existing support worker sessions with clients. Group B comprised home care clients with dementia/cognitive impairment and family carer dyads. The inclusion criteria comprised home care clients who were (i) aged ≥ 60 years; (ii) medically diagnosed with dementia, or those having a cognitive score of ≤22 on the Rowland Universal Dementia Assessment Scale (RUDAS) [13] or receiving a dementia and cognition supplement for home care; (iii) able to walk independently or with supervision (with or without a walking aid); (iv) approved by their medical doctor to participate in exercises if they answered “yes” to any of the questions in the Physical Activity Readiness Questionnaire [14]; (v) willing to perform exercises; (vi) residing in their own homes in Melbourne, Australia; and (vii) cared for by a family carer who was able to supervise the exercises three days per week. Family carers were those who could supervise the person with dementia/cognitive impairment for exercises or able to share supervision with care support workers to meet the weekly exercise target. 

Home care clients were excluded if they were (i) participating in another regular exercise program (≥2 days/week and >30 min each day) or (ii) planning a holiday or residential care respite for more than two weeks in the 12 weeks ahead, including their family carers.

In Australia, care support workers typically receive vocational training to deliver home care services, e.g., personal and domestic care to older people to support their living at home. There is no minimum requirement for these workers to be trained in how to promote physical activity in their clients. For this study, care support workers were recruited from community aged care provider organisations in Melbourne, Australia, through the promotion of the training course and nominations by managers in the participating organisations. Home care clients with dementia/cognitive impairment and family carers were recruited by the managers of the participating organisations.

The study was approved by the Human Research Ethics Committees of Monash University (ID 28838). Consent was obtained from all participants and, if required, their responsible person (for the person with dementia) prior to baseline assessments.

### 2.3. Baseline Assessments

Home care clients and family carers were interviewed at home by the chief researcher. Baseline (T_0_) data were collected for the clients’ demographic information including culturally and linguistically diverse (CALD) background, presence of medical diagnosis of dementia and/or other medical conditions, RUDAS cognitive score, years living with diagnosed dementia, dementia type, level of home care package, carer-reports on client’s physical capacity to ambulate inside and outside their home, ability to walk up and down three stairs without a handrail, and bend to pick up an object from the floor.

Through Government-funded programs, home care package levels are assessed according to the needs of older people. Level 4 home care package are provided for those who have high care needs, level 3 for intermediate care needs and level 2 for low care needs. Participants were classified as being from a CALD background if they were born outside of Australia and spoke a language other than English at home [15].

#### 2.3.1. Safety and Adherence

Care support workers, clients and family carers were provided with diaries and instructions to record the exercises, the duration and days of exercise performance, if adverse events including falls occurred during the 12-week program, and the reasons if not meeting the number of days for the exercise target. In addition, care support workers could enter in their diaries if any care was missed and extra time spent outside of their care shift because of the exercise program. Any alternate physical activity performed in place of the exercise program when a client did not wish to perform the exercises was also recorded. 

#### 2.3.2. Potential for Benefits 

Validated scales and tests were used to assess various client outcomes at baseline (T_0_). The Phone-FITT scale [16] (frequency + duration FD summary score) was used to measure household and recreational activities, with higher total FD scores indicating greater participation in that category of physical activity. The KATZ Activities of Daily Living Index [17] was used to assess a client’s independence to perform six daily activities of living. A summative score ranged from 0–6, in which a score of 6 indicated full function, 4 indicated moderate impairment and ≤2 indicated severe functional impairment. The Iconographic Falls Efficacy Scale [18] was used to assess client’s fear of falling by using ten pictures to describe a range of activities and situations. A summative score ranged from 10–40 in which a score of 10–18 indicated low concern about falling and 19–40 indicated high concern about falling. The QoL-AD [19] has 13 items that measure quality of life in people with dementia. A summative score is in the range of 13–52, in which a higher score indicated a better quality of life. The functional mobility of clients was assessed using the Time-Up-and-Go test [20] at a maximal but safe speed, whereby a time ≥12 s indicated the person was at risk of falling. Lower limb functioning was evaluated using the Short Physical Performance Battery test [21] which consisted of balance tests (side-by-side stand, semi-tandem stand and tandem stand), a gait speed test (time to walk a 3-metre path, with the use of a walking aid if needed), and a chair-stand test (time to complete 5 chair stands). Each test category scored 0–4, whereby a higher score indicated better standing balance, higher gait speed and better lower limb strength, respectively. The summative score of all test categories (sum of balance test, gait speed test and chair stand test scores) ranged from 0–12, whereby a score of 12 indicated full function and <10 indicated one or more mobility limitation.

Client and/or carer rated client’s sleep quality in the past three months at baseline (T_0_) on a 5-point Likert scale. A rating of “1” represented “very good” sleep quality, “3” represented “average” and “5” represented “very poor” sleep quality. Data on carer- and/or client-reported falls in the past month, whereby a fall was defined as “an event which results in a person coming to rest inadvertently on the ground or floor or other lower level” [22], and healthcare use in the past three months were also collected.

### 2.4. Intervention

The physical exercise program, entitled “*Safe Functional Home Exercise” program for people with dementia*, was co-designed by an expert advisory panel comprising seven exercise and falls prevention researchers from Melbourne, Sydney and Perth of Australia, two clinical physiotherapists, five community aged care industry and consumer stakeholders from Melbourne, Australia, through two 2 h meetings scheduled one month apart. At the first meeting, the panellists co-designed an approach that consisted of a physical activity program and exercise choices to guide the development of an exercise program with inputs from (a) panellists, contributing their perspectives; (b) elements of current exercise programs that are safe and effective for older people to improve their strength, balance and fitness (identified by the research team through a desktop review); and (c) implementation principles (client and organisational needs, program design with low complexity and costs) to suit home care clients with dementia. After this meeting, the research team synthesised these inputs and discussion to draft an exercise program incorporating exercise choices and progression, and the associated staff training resources. At the second meeting, the exercises and staff training resources were reviewed and refined. After the second meeting, an exercise protocol, staff training program and resources were developed that included safety aspects (precautions/identification of red flags) of delivering exercises by care support workers (and family carers).

The “*Safe Functional Home Exercise” program for people with dementia* was a 12-week program that used a simplified approach with a set of four exercises. Briefly, Exercise A is knee squats (knee strengthening exercise), Exercise B is heel raises (ankle strengthening exercise), Exercise C is a stepping and balance exercise (body weight shift and balance training exercise) and Exercise D is a dual task of walk and talk exercise (concurrent mobility and cognitive task to train dual task ability). Each exercise had two levels: Level 1 (easy) exercises were performed with hand support (Figure 1), while Level 2 (moderate) exercises were performed without hand support (Figure 2). More photos for level 1 and 2 exercises A, B, C and D are shown in Appendix A.

Exercises A, B and C could be progressed to level 2 at any time during the 12-week program when the client could safely complete two sets of 8–12 repetitions of a level 1 exercise with no more than one minute rest in between the sets. Exercise D could be progressed to level 2 when the client could perform level 1 “talking task” with no error and was steady in walking. The difficulty of the exercises was able to be increased for exercise A, B or C at level 1 or 2 by reducing the speed of exercise, base of feet support, hand support (for level 1 exercise only) and holding the end of movement for 5 s or more time, and for Exercise D by doing a more challenging talking task, to ensure clients were exercising at their optimal capacity. All clients commenced the exercise program at level 1 at week 1. 

All participating care support workers were trained by the chief researcher, who is also a registered physiotherapist in Australia, with over 15 years of aged care experience (PhD, Graduate Diploma (Occupational and Environmental Health), Bachelor of Applied Science (Physiotherapy)). The 2-half-day training course utilised a face-to-face and video-conferencing format, aiming to provide staff with the knowledge, skills and confidence to deliver the exercise program. It also included safety aspects, ways to engage clients, exercise progression and maintenance, explanation of program documents (exercise manual, home exercise templates and exercise diary), and red flag events to alert the chief researcher and client managers. The exercise target for care support workers was 15 min once weekly for 12 weeks during their care shifts. They would invite family carers to observe the exercises and provide them with home exercise templates (pre-made by the research team) to supervise exercises in between their visits. 

The chief researcher explained the exercise program to clients and family carers, and provided them with program documents (exercise manual and exercise diary) at the initial home visit. Instructions were given for safety and how to complete the exercise diary. The exercise target for family carers was 30 min each day, three days per week for 12 weeks, i.e., several times per week in addition to those performed during the care support worker’s shifts. They would observe the exercises and followed the home exercise templates provided by care support workers each week to supervise exercises. The exercises could be performed in smaller blocks of time throughout the day to reduce client fatigue. They were able to share their supervision load with care support workers to meet their exercise target. For example, if the care support worker assisted for one day per week (beyond their one day per week requirement), family carers would supervise clients to exercise two days per week.

The chief researcher provided support to care support workers and client/family carer dyads every fortnight via telephone contact during the 12-week program to provide motivation, help with troubleshooting and ensure safety, exercise progression and maintenance. Care support workers and client/family carer dyads could also contact the chief researcher at any time if there were questions. The manual for the *“Safe Functional Home Exercise” program for people with dementia* is shown in Appendix A. A short refresher video for the staff training course can be viewed on YouTube (https://youtu.be/xxYQtns9in0).

### 2.5. Follow-Up Assessments

#### 2.5.1. Safety and Adherence

Care support workers and family carers sent photos of their exercise dairies for the past four weeks via the Whatsapp or WeChat to the chief researcher at Week 4 (T_1_), Week 8 (T2) and Week 12 (T_3_). A follow-up was made to ensure all exercise diaries were received. 

#### 2.5.2. Feasibility

The recruitment of home care client/family carer dyads and dropout rates for all participants were calculated at the point of project completion. The acceptance and confidence of care support workers and client/family carer dyads for the exercise program were evaluated using a separate survey (not included in this paper; planned for a future manuscript). 

#### 2.5.3. Potential for Benefits

Validated scales and tests were used to reassess client outcomes at Week 12 (T_3_) via a second home visit by the chief researcher. The Phone-FITT scale, the KATZ index of independence in activities of daily living, the Iconographic Falls Efficacy scale, the QoL-AD, the Timed-Up-and-Go test, the Short Physical Performance Battery test and the Likert scale of sleep quality in the past 3 months were repeated. Data for carer and/or client reports of healthcare use in the past three months and falls in the past month were collected. In addition, the Phone-FITT scale measures falls, and reports of falls in the past month were also collected at Week 4 (T_1_) and Week 8 (T_2_) via telephone contacts. A schema of the implementation and assessment of the “Safe Functional Home Exercise” program for people with dementia is shown in Figure 3.

### 2.6. Data Analysis 

The data were analysed with STATA SE version 15.1 (StataCorp. 2017). The adherence rate by exercise time and number of exercise days per week supervised by care support workers and family carers were calculated at T_1_, T_2_ and T_3_ for the past 4 weeks for clients that remained in the program at each time-point, and for all the time-points combined against the exercise target for each group. The recruitment rate against the project target of 30 home care client/family carer dyads and study dropout rates were calculated. The safety associated with the program implementation was also described.

The potential benefits of the exercise intervention on client outcomes were compared between outcome (dependent) variables at baseline (T_0_) and Week 12 (T_3_) with independent variables (time-point) coded as 0 for T_0_ and 1 for T_3_. Linear regression analysis with data clustered by individual participants utilising robust standard error [23] was employed to examine the differences between continuous outcome variables, in addition to ordinal logistic regression for ranked outcome variables pre- and post-intervention. Participants who attempted but were physically unable or cognitively unable to comprehend and execute the requirements of the assessments were allocated a score of “999” (not missing data) and included in the regression analyses. For this purpose, the outcome variable that contained data coded as “999” was treated as ranked data and analysed using ordinal logistic regression analysis but excluded from the calculation of mean and SD for that variable. The differences between program completers and non-completers with variables at baseline (T_0_) were compared using logistic regression analyses, with the dependent variable (completers) coded as 1 and (non-completers) coded as 0. All statistical analyses were conducted with the alpha criterion set at *p* < 0.05.

## 3. Results

A total of 38 potential home care client/family carer dyads from two aged care organisations were assessed for eligibility. Of these, 12 were excluded due to refusal for different reasons and not meeting inclusion criteria. Participant flow through the study is shown in Figure 4. 

Twenty-six client/family carer dyads (Table 1) and 26 care support workers commenced the exercise intervention. Among the home care clients, 14 had diagnosed dementia and 12 had cognitive impairment (based on RUDAS but no formal diagnosis). Fifteen (57.7%) were female; mean age was 84.5 years (SD 6.5) and 21 (80.8%) were from a CALD background. The mean RUDAS score was 16.9 (SD 4.7), with 17 (65.4%) classified as having mild dementia (RUDAS score 17–22), 7 (26.9%) moderate (RUDAS score 10–16) and 2 (7.7%) severe dementia (RUDAS score <10). Alzheimer’s disease was the most common dementia type, accounting for 83.3% of participants with known diagnosis. All clients were receiving a home care package, with 42.3% being level 3 (intermediate care needs) and 42.3% level 4 (high care needs) recipients. Most clients had difficulty moving around their home (54.6%) and outside their home (80.8%), negotiating stairs (92.3%) and bending to pick up an object from the floor (53.8%). Fifteen client/family carer dyads (57.7%) completed the exercise program (Figure 3). There were no significant differences identified in baseline variables between program completers and non-completers (Table 1). Of the program completers, only one remained at Level 1 for exercise A, B, C and D throughout the program, while 14 progressed from level 1 to level 2 exercises. Of the clients who progressed, seven progressed to level 2 for all exercises, and seven progressed to level 2 in one to three exercises, all with the difficulty of exercise increased.

### 3.1. Safety and Adherence

No adverse/serious events or falls occurred during the exercises supervised by care support workers or family carers. There were no missed care requirements of the care package reported during care shifts and no extra time was spent outside the scheduled care shifts because of the exercise program. Two clients reported aggravation of arthritic knee pain during performance of exercise A (knee squats). Care support workers and family carers were instructed by the chief researcher to modify exercise A (e.g., by reducing the range of movement and repetition of exercise and using hand support when performing the exercise or changing to a seated position to perform knee extension exercises instead). An episode of near miss falling during family carer’s supervision of exercises was reported and prevented by the family carer. Walking outdoors was the alternate physical activity performed to substitute some of the exercises in the program when six clients did not wish to perform the exercises on certain days (used as an option in preference to no exercise).

There was an apparent drop-off for the total time of exercise and total days of exercise in the past four weeks at Week 4, 8 and 12 over the period (Table 2), but this was due to the withdrawal of client/family carer dyads at different times in the program (Figure 3), and hence, increasingly fewer participants remained in the program at each time-point for the adherence measures to be taken. Overall, care support workers and family carers had excellent adherence to the exercise program for all time-points. The average time supervised by care support workers and family carers each week for all time-points was 20.6 min and 73.9 min, respectively (Table 2). The average days of exercise supervised by care support workers and family carers each week for all time time-points were 0.8 days and 3.1 days, respectively (Table 2). The adherence rates against the exercise time target by care support workers and family carers across all time-points were 137.3% and 82.1%, respectively, while the adherence rates for exercise days by care support workers and family carers were 79.6% and 104.8%, respectively (Table 2). About 52% of family carers were assisted by care support workers to reach their exercise target across all time-points (Table 2). Common reasons for not exercising included clients/family carers/care support workers cancelling care shifts because of COVID-19 infection or being a close contact with COVID-19 patients needing isolation, medical appointments, being unwell, clients/family carers going on unplanned trips or family carers were too busy.

### 3.2. Feasibility

The recruitment rate of home care client/family carer dyads over the eight-month recruitment period was 86.7%. No care support workers dropped out from the exercise program. Eleven client/family carer dyads withdrew, resulting in a dropout rate of 42.3%. Five clients became medically unwell, three were unable to engage in exercise, one changed aged care provider, one whose carer had increased carer burden and one moved to a nursing home (Figure 3). 

### 3.3. Potential for Benefits 

Physical activity participation, physical function and falls efficacy significantly improved at Week 12 compared to baseline (Table 3). Physical activity participation increased significantly among home care clients [coefficient (robust 95% CI): 19.31 (13.67, 24.95)], and in the subscales of household [8.33 (4.32, 12.35)] and recreational [10.98 (7.67, 14.29)] activities. Lower limb function improved significantly [1.33 (0.23, 2.43)], with better standing balance [0.6 (0.13, 1.07)] demonstrated. Falls efficacy also showed a significant improvement [−4.93 (−6.94, −2.93)] at Week 12. There were no significant differences identified in other outcomes examined.

## 4. Discussion

To the best of our knowledge, this is the first study to have tested the feasibility, safety and potential benefits of utilising trained care support workers to deliver a co-designed physical exercise program for people with dementia or cognitive impairment and their family carers. The feasibility of the method was demonstrated by the recruitment of home care client/family carer dyads and care support workers in the study. However, there was a high dropout rate of 42% for client/family carer dyads in the exercise program due to different reasons. The safety of the exercise program was demonstrated, and care support workers were trained to deliver this 12-week exercise program to home care clients with dementia or cognitive impairment, combined with regular remote support from a physiotherapist. The adherence to the exercise program was excellent from both care support workers and family carers. Physical activity participation, physical function and falls efficacy significantly improved at Week 12 compared to baseline, indicating the potential benefits of this co-designed exercise program regarding these outcomes. There were no differences identified in healthcare use and fall outcomes.

This study demonstrated that the co-designed physical activity program is feasible and can be implemented safely by trained care support workers in home care services and family carers, and may achieve good outcomes for people with dementia [24,25,26]. Physical activity programs have previously been safely implemented for people with dementia, often with carer support [27]. Our study demonstrated the *“Safe Functional Home Exercise” program for people with dementia*, which is a multi-modal exercise program that consists of leg strengthening, balance and mobility training, and has the potential to improve physical activity participation, physical function and falls efficacy among people with dementia or cognitive impairment. This finding is consistent with a systematic review of 43 clinical trials which showed that supervised multimodal exercises for 2–3 days a week can improve physical function in people with various levels of cognitive impairment [27], but the evidence on falls is inconclusive. Unlike “off-the-shelf” exercise programs, e.g., the Otago exercise program [28], which was often used by researchers and practitioners for older people who had intact cognition, the *“Safe Functional Home Exercise” program for people with dementia* was co-designed specifically for older people with dementia, using inputs from the aged care industry stakeholders and consumers. Larger trials with a control group design are needed to evaluate the effectiveness of this co-designed exercise program and the method of exercise delivery, i.e., through trained care support workers partnering with family carers, and using a physiotherapist for program support.

The use of an existing workforce of care support workers to facilitate older people with dementia or cognitive impairment to participate in physical activity during home care visits is scalable and creates an opportunity to capitalise on the existing aged care system. Indeed, our study showed that no missed care occurred and no extra time was spent outside the scheduled care shifts during the 12-week program. This meant that the cost of exercise intervention alone was covered by the home care package funds, without clients having to spend out of pocket to participate in the exercises. Thus, it has the capacity to benefit many home care clients with dementia. However, the additional cost of using a physiotherapist to train care support workers and provide intermittent support need to be considered in real-life scenarios, and that these costs may need to be included in the packaged funds, as well as payment for staff time to attend training sessions and produce training resource materials. It is important to train care support workers not only to equip them with the knowledge and skills needed to deliver a physical activity program to clients, but also from a safety and confidence perspective. Although our training appears to be sufficient in terms of the safe delivery of the program (by care support workers and family carers), future studies may explore care support workers’ confidence in this regard, and whether there are aspects of the training that can be modified to meet any perceived areas of need. In addition, further studies are required to determine the best model for sustaining physical activity participation, for example, whether the exercise program can be continued by care support workers beyond a 12-week period without physiotherapy support, or whether this model is better offered to home care clients with dementia with physiotherapy support as a 12-week burst from time to time, as considered necessary by the aged care organisations. 

Over 80% of home care client/carer dyads participating in this study were from a CALD background. These populations often find it difficult to access mainstream physical activity programs, as they are more likely to become home-bound or encounter cultural and language barriers in accessing suitable programs [29,30]. In our study, all care support workers of CALD clients were from the same CALD background and could speak the same language. Therefore, this method of exercise intervention could provide a potential way for people with dementia or cognitive impairment from CALD communities to perform physical activity. Co-designing tailored approaches and involving the community to facilitate physical activity have previously been shown to improve physical activity adherence [31]. This is consistent with the excellent adherence rates for all time-points achieved by care support workers and family carers in our study. Although the *“Safe Functional Home Exercise” program for people with dementia* was co-designed for people with dementia, it was not co-designed with a CALD focus. Future research aiming to refine or modify the co-designed exercise program to suit CALD groups could support people with dementia including those from CALD backgrounds to participate in physical activity.

There are strengths and limitations that need to be considered. The participants were recruited from two aged care organisations in Victoria; this increased the generalisability of the results. The small sample size in our feasibility study lacked statistical power to detect the effectiveness of the co-designed exercise program on some important outcomes, but the preliminary results on some outcomes of interest are promising. Our study showed excellent adherence to the exercise requirement, i.e.,105 min per week for four days per week (15 min one day/week for care support workers and 90 min three days/week for client/family carer dyads); however, this is below the current physical activity guidelines for older people, which recommends 150 min per week of exercise, performed across most days of the week [32]. Additional physical activity opportunities are needed for people with dementia or cognitive impairment including those from a CALD background to meet the physical activity guidelines for promoting good health and wellbeing. 

Future large studies with an experimental design are needed in order to evaluate the effectiveness of the *“Safe Functional Home Exercise” program for people with* dementia. Strategies to minimise potentially avoidable reasons for dropouts such as providing practical strategies for care support workers and family carers to engage clients with dementia or cognitive impairment to perform exercise, and supporting family carers to reduce carer burden in future studies are required.

## 5. Conclusions

Feasibility, safety, good adherence and potential for benefits in physical activity participation, physical function and falls efficacy were demonstrated for the co-designed exercise program “*Safe Functional Home Exercise” program for people with dementia*. The method of exercise delivery by trained care support workers in partnership with family carers and physiotherapy support is potentially scalable in the home care sector. Future randomised controlled trials of the exercise program with an embedded evaluation of the implementation outcomes are required. Strategies to minimise dropouts in future effectiveness studies are also required.

## Figures and Tables

**Figure 1 ijerph-20-03662-f001:**
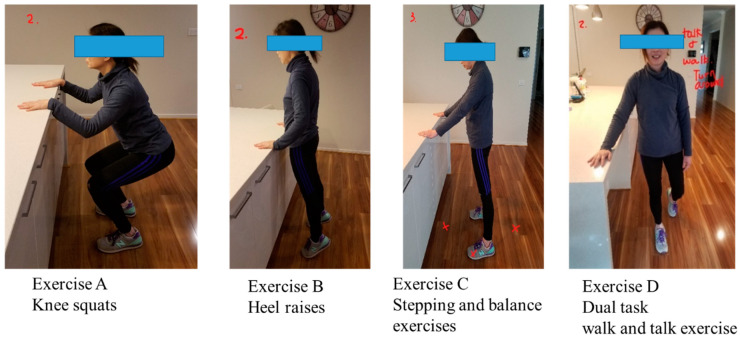
“Safe Functional Home Exercise” program for people with dementia (level 1 exercises).

**Figure 2 ijerph-20-03662-f002:**
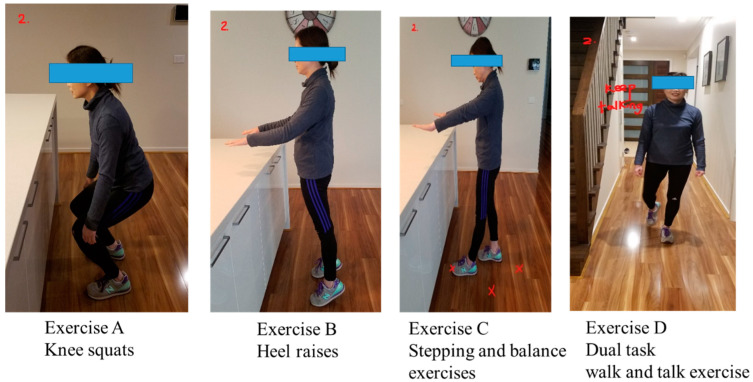
“Safe Functional Home Exercise” program for people with dementia (level 2 exercises).

**Figure 3 ijerph-20-03662-f003:**
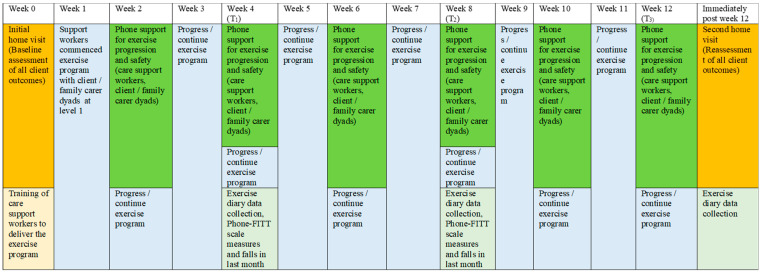
A schema of the implementation and assessment of the “Safe Functional Home Exercise” program.

**Figure 4 ijerph-20-03662-f004:**
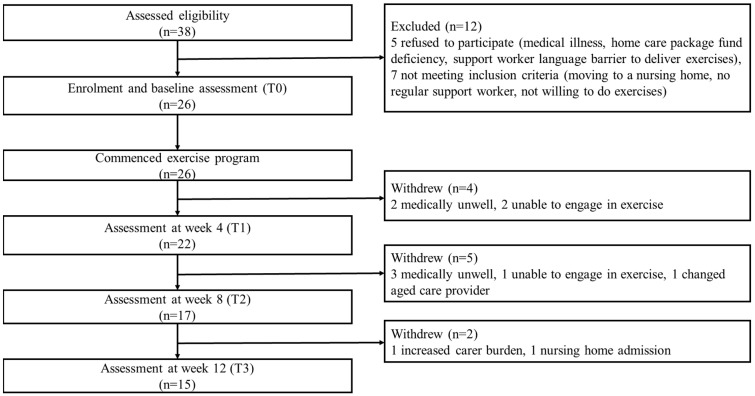
Participant flow chart (home care client/family carer dyads).

**Table 1 ijerph-20-03662-t001:** Participant demographics/characteristics at baseline (T_0_) and statistical comparisons between program completers and non-completers at T_0_.

	Baseline Data (T_0_) of All Participants (*n* = 26)	Baseline Data (T_0_) of Program Completers (*n* = 15)	Baseline Data (T_0_) of Non-Completers (*n* = 11)	Coefficient (95% CI) ^a^	*p*-Value
**Age (years)—mean (SD)**	84.5 (6.5)	83.5 (6.8)	85.8 (6.1)	−0.06 (−0.19, 0.07)	0.36
**Gender (female)—*n* (%)**	15 (57.7)	10 (66.7)	5 (45.5)	0.88 (−0.72, 2.48)	0.28
**Lives alone—*n* (%)**	4 (15.4)	3 (20)	1 (9.1)	0.92 (−1.50, 3.33)	0.46
**Medically diagnosed dementia—*n* (%)**	14 (53.9)	7 (46.7)	7 (63.6)	−0.69 (−2.29, 0.90)	0.39
**RUDAS score ^b^—mean (SD)**	16.9 (4.7)	18.1 (3.7)	15.36 (5.55)	0.14 (−0.05, 0.33)	0.15
**Years since dementia diagnosis—mean (SD)**	4.3 (2.9)	4.4 (2.8)	4.2 (3.2)	0.04 (−0.34, 0.41)	0.85
**Type of dementia if diagnosed and known—*n* (%)**					
Alzheimer’s disease	10 (83.3)	5 (83.3)	5 (83.3)		N/A
Vascular dementia	0	0	0		
Lewy body dementia	0	0	0		
Mixed dementia	2 (16.7)	1 (16.7)	1 (16.7)		N/A
**Home care package level ^c^—*n* (%)**				−0.01 (−1.11, 1.08)	0.98
Level 4	11 (42.3)	5 (33.3)	6 (54.6)		
Level 3	11 (42.3)	9 (60.0)	2 (18.2)		
Level 2	4 (15.4)	1 (6.7)	3 (27.3)		
**Received dementia/cognition supplement—*n* (%)**	8 (30.8)	4 (26.7)	4 (36.4)	−0.45 (−2.13, 1.23)	0.60
**CALD status ^d^—*n* (%)**	21 (80.8)	12 (80.0)	9 (81.8)	−0.12 (−2.10, 1.87)	0.91
**Medical history (apart from dementia/cognitive impairment)—*n* (%)**					
Hypertension	18 (69.2)	10 (66.7)	8 (72.7)	−0.29 (−1.99, 1.42)	0.74
Depression	16 (61.5)	10 (66.7)	6 (54.6)	0.51 (−1.09, 2.11)	0.53
Arthritis	12 (46.2)	7 (46.7)	5 (45.5)	0.05 (−1.51, 1.61)	0.95
Anxiety	11 (42.3)	4 (26.7)	7 (63.6)	−1.57 (−3.25, 0.11)	0.07
Hearing loss	11 (42.3)	6 (40.0)	5 (45.5)	−0.22 (−1.80, 1.35)	0.78
Diabetes	8 (30.8)	5 (33.3)	3 (27.3)	0.29 (−1.42, 1.99)	0.74
Heart disease	8 (30.8)	5 (33.3)	3 (27.3)	0.29 (−1.42, 1.99)	0.74
Broken bones	8 (30.8)	6 (40.0)	2 (18.2)	1.10 (−0.75, 2.95)	0.24
Stroke	6 (23.1)	3 (20.0)	3 (27.3)	−0.41 (−2.24, 1.43)	0.67
Lung disease/respiratory disease	5 (19.2)	3 (20.0)	2 (18.2)	0.12 (−1.87, 2.10)	0.91
Cancer	5 (19.2)	3 (20.0)	2 (18.2)	0.12 (−1.87, 2.10)	0.91
Visual impairment not correctable by glasses	4 (15.4)	3 (20.0)	1 (9.1)	0.92 (−1.50, 3.33)	0.46
Hip joint replacement	3 (11.5)	3 (20.0)	0	0	N/A
Kidney disease/renal failure	2 (7.7)	1 (6.7)	1 (9.1)	−0.34 (−3.22, 2.55)	0.82
Parkinson’s Disease	2 (7.7)	2 (13.3)	0	0	N/A
Knee joint replacement	1 (3.9)	1 (6.7)	0	0	N/A
Other (osteoporosis, laparotomy)	14 (53.9)	8 (53.3)	6 (54.6)	−0.05 (−1.61, 1.51)	0.95
**Carer-reported physical capacity—*n* (%)**				−0.12 (−1.51, 1.27)	0.87
**Move around home**					
Can perform without difficulty	5 (45.5)	6 (40.0)	5 (45.5)		
Can perform but with difficulty	5 (45.5)	9 (60.0)	5 (45.5)		
Cannot perform without supervision from someone else	1 (9.1)	0	1 (9.1)		
**Carer-reported physical capacity—*n* (%)**				0.29 (−0.87, 1.46)	0.62
**Move outside home**					
Can perform without difficulty	5 (19.2)	2 (13.3)	3 (27.3)		
Can perform but with difficulty	14 (53.9)	9 (60.0)	5 (45.5)		
Cannot perform without assistance from someone else	7 (26.9)	4 (26.7)	3 (27.3)		
**Carer-reported physical capacity—*n* (%)**				0.70 (−0.65, 2.05)	0.31
**Walk 3 stairs without a handrail or assistance from someone else**					
Can perform without difficulty	2 (7.7)	1 (6.7)	1 (9.1)		
Can perform but with difficulty	4 (15.4)	1 (6.7)	3 (27.3)		
Cannot perform without assistance from someone else	20 (76.9)	13 (86.7)	7 (63.6)		
**Carer-reported physical capacity—*n* (%)**				0.44 (−0.53, 1.41)	0.37
**Bend and pick up an object from the floor without assistance from someone else**					
Can perform without difficulty	12 (46.2)	6 (40.0)	6 (54.6)		
Can perform but with difficulty	7 (26.9)	4 (26.7)	3 (27.3)		
Cannot perform without assistance from someone else	7 (26.9)	5 (33.3)	2 (18.2)		
**Number of fall(s) in the past month—*n* (%)**				0.40 (−0.56, 1.37)	0.41
0	20 (76.9)	11 (73.3)	9 (81.8)		
1	2 (7.7)	1 (6.7)	1 (9.1)		
≥1	4 (15.4)	3 (20.0)	1 (9.1)		
**Healthcare use in the past 3 months—*n* (%)**					
Admitted to the emergency department	2 (7.7)	0	2 (18.2)	0	N/A
Admitted to hospital	2 (7.7)	2 (13.3)	0	0	N/A
Consulted general practitioner	24 (92.3)	14 (93.3)	10 (90.9)	0.34 (−2.55, 3.22)	0.82
Consulted other health professional(s)	11 (42.3)	5 (33.3)	6 (54.6)	−0.88 (−2.48, 0.72)	0.28
**Phone-FITT FD score—mean (SD)**					
Household FD score	10.7 (10.1)	11.0 (8.8)	10.2 (12.2)	0.01 (−0.07, 0.09)	0.84
Recreational FD score	7.2 (5.2)	6.8 (4.8)	7.8 (5.9)	−0.04 (−0.20, 0.11)	0.60
Total FD score	17.9 (10.6)	17.8 (9.3)	18.0 (12.6)	−0.00 (−0.08, 0.07)	0.95
**Timed-Up-and-Go test (seconds)—mean (SD)**	21.2 (13.5)	19.1 (6.1)	24.4 (20.3) ^e^	−0.03 (−0.09, 0.03)	0.35
**Short Physical Performance Battery—mean (SD)**					
Balance tests score	2.2 (1.3)	2 (1.1)	2.6 (1.6)	−0.33(−0.95, 0.29)	0.30
Gait speed Test score	2.3 (1.2)	2.3 (0.8)	2.3 (1.6)	0.05 (−0.64, 0.73)	0.89
Chair-stand test score	1.6 (1.5)	1.3 (1.4)	1.9 (1.6)	−0.29 (−0.85, 0.27)	0.31
Total score	6.1 (3.4)	5.7 (2.7)	6.7 (4.2)	−0.10 (−0.34, 0.14)	0.42
**KATZ Activities of Daily Living Index—mean (SD)**	4.7 (1.2)	4.7 (1.1)	4.7 (1.3)	−0.04 (−0.71, 0.62)	0.90
**Iconographic Falls Efficacy Scale—mean (SD)**	29.7 (7.8)	31.0 (6.5)	28.0 (9.4)	0.05 (−0.05, 0.16)	0.33
**Quality of Life-AD—mean (SD)**					
Client-rated	31.5 (6.3)	32.2 (6.8)	30.5 (5.7)	0.05 (−0.08, 0.18)	0.48
Carer-rated	28.2 (5.0)	28.8 (5.4)	27.4 (4.6)	0.06 (−0.10, 0.22)	0.47
**Sleep quality rating in the past 3 months—mean (SD)**	3.0 (1.2)	3.1 (1.2)	2.8 (1.1)	0.20 (−0.51, 0.91)	0.58

^a^ Based on logistic regression analyses. Comparisons were made between program completers and non-completers with variables at baseline (T_0_) and dependent variable (completers) coded as 1 and (non-completers) coded as 0. ^b^ Rowland Universal Dementia Assessment Scale (RUDAS). Scores ranged from 0–30, with scores 23–30 considered as normal and score 22 or less (lower score indicated greater cognitive impairment) needed to be considered in the clinical context. ^c^ Level 4 indicated high care needs, level 3 intermediate care needs and level 2 low care needs. ^d^ Participants were from culturally and linguistically diverse (CALD) background if they were born overseas and spoke a language other than English at home. ^e^ One participant was unable to perform the Timed-Up-and-Go test and was not included in the calculation of mean and SD. This missing data was coded as “999” and was included in the statistical comparisons between program completers and non-completers at T_0_ using ordinal logistic regression.

**Table 2 ijerph-20-03662-t002:** Exercise time, exercise days and adherence to exercise program by care support workers and family carers during the 12-week program.

	Week 4 (T_1_) *n* = 22	Week 8 (T_2_) *n* = 17	Week 12 (T_3_) *n* = 15	All Time-Points (T_1_ + T_2_ + T_3_)
**Total time (minutes) of exercise program supervised by care support workers in the past 4 weeks ^a^**	1846	1511	1090	4447
**Average time (minutes) supervised by care support workers each week**	21	22.2	18.2	20.6
**Adherence rate (%) by exercise time supervised by care support workers ^b^**	139.9	148.1	121.1	137.3
**Total time (minutes) of exercise program supervised by family carers in the past 4 weeks ^c^**	6067	5240	4656	15,963
**Average time (minutes) supervised by family carers each week**	68.9	77.1	77.6	73.9
**Adherence rate (%) by exercise time supervised by family carers ^d^**	76.6	85.6	86.2	82.1
**Total days of exercise supervised by care support workers in the past 4 weeks ^e^**	71	58	43	172
**Average days of exercise supervised by care support workers each week**	0.8	0.9	0.7	0.8
**Adherence rate (%) by days of exercise supervised by care support workers ^f^**	80.7	85.3	71.7	79.6
**Total days of exercise supervised by family carers in the past 4 weeks ^g^**	260	220	199	679
**Average days of exercise supervised by family carers each week**	3.0	3.2	3.3	3.1
**Adherence rate (%) by days of exercise supervised by family carers ^h^**	98.5	107.8	110.6	104.8
**Number of family carers assisted by care support workers to reach their exercise target ^i^—*n* (%)**	12 (54.6)	9 (52.9)	7 (46.7)	28 (51.9)

^a^ The minimum target of exercise time supervised by care support workers was 1 day/week for 15 min for 12 weeks. The minimum target required at T_1_, T_2_, T_3_ for the past 4 weeks = *n* (clients remained at that time point for data to be collected) × 15 min × 4 weeks (i.e., 1320 min at T_1_, 1020 min at T_2_ and 900 min at T_3_). ^b^ Time/1320 min × 100% for T_1_, Time/1020 min × 100% for T_2_, Time/900 min × 100% for T_3_, Time/(1320 + 1020 + 900 min) × 100% for all time-points. ^c^ The minimum target of exercise time supervised by family carers was 3 days/week, each day for 30 min for 12 weeks. The minimum target required at T_1_, T_2_, T_3_ for the past 4 weeks = *n* (clients remained at that time point for data to be collected) × 90 min × 4 weeks (i.e., 7920 at T_1_, 6120 min at T_2_ and 5400 min at T_3_). ^d^ Time/7920 min × 100% for T_1_, Time/6120 min × 100% for T_2_, Time/5400 min × 100% for T_3,_ Time/(7920 + 6120 + 5400 min) × 100% for all time-points. ^e^ The minimum target of exercise days supervised by care support workers was 1 day/week for 12 weeks. The minimum target required at T_1_, T_2_, T_3_ for the past 4 weeks = *n* (clients remained at that time point for data to be collected) × 1 day per week × 4 weeks (i.e., at 88 days at T_1_, 68 days at T_2_ and 60 days at T_3_). ^f^ Days/88 days × 100% for T1, Days/68 days × 100% for T2, Days/60 days × 100% for T3, Days/(88 + 68 + 60 days) × 100% for all time-points. ^g^ The minimum target of exercise days supervised by family carers was 3 days/week for 12 weeks. The minimum target required at T_1_, T_2_, T_3_ for the past 4 weeks = *n* (clients remained at that time point for data to be collected) × 3 days/week × 4 weeks (i.e., at 264 days at T_1_, 204 days at T_2_ and 180 days at T_3_). ^h^ Days/264 days × 100% for T1, Days/204 days × 100% for T2, Days/180 days × 100% for T3, Days/(264 + 204 + 180 days) × 100% for all time-points. ^i^ Care support workers provided assistance to family carers to help them reach their exercise target. Assistance ranged from 1–3 days/week, i.e., if care support worker assisted one day/week, family carers would supervise clients to exercise two days/week to reach their target.

**Table 3 ijerph-20-03662-t003:** Baseline data (T_0_), Week 4 data (T_1_), Week 8 data (T_2_) and Week 12 data (T_3_) of program completers, and effect of the exercise program on outcome variables for program completers.

	Baseline Data (T_0_) of Program Completers (*n* = 15)	Week 4 Data (T_1_) of Program Completers (*n* = 15)	Week 8 Data (T_2_) of Program Completers (*n* = 15)	Week 12 Data (T_3_) of Program Completers (*n* = 15)	Co-Efficient (Robust 95% CI) ^a^	*p*-Value
**Phone-FITT FD score—mean (SD)**						
Household FD score	11 (8.77)	13.27 (9.58)	18.07 (12.20)	19.33 (10.51)	8.33 (4.32, 12.35)	<0.01 *
Recreational FD score	6.75 (4.79)	15.14 (2.49)	18.48 (4.67)	17.73 (4.70)	10.98 (7.67, 14.29)	<0.01 *
Total FD score	17.75 (9.29)	28.08 (10.16)	36.55 (10.43)	37.06 (10.14)	19.31 (13.67, 24.95)	<0.01 *
**Timed-Up-and-Go test(s)—mean (SD)**	19.14 (6.14)	--	--	18.10 (7.85)	−1.04 (−5.37, 3.30)	0.62
**Short Physical Performance Battery—mean (SD)**						
Balance test score	2 (1.13)	--	--	2.6 (1.06)	0.6 (0.13, 1.07)	0.02 *
Gait speed Test score	2.33 (0.82)			2.67 (1.11)	0.33 (−0.13, 0.79)	0.14
Chair stand test score	1.33 (1.35)			1.73 (1.39)	0.40 (−0.27, 1.07)	0.22
Total score	5.67 (2.66)			7 (2.85)	1.33 (0.23, 2.43)	0.02 *
**KATZ Index of Independence in Activities of Daily Living—mean (SD)**	4.67 (1.11)	--	--	4.67 (1.50)	0 (−0.30, 0.30)	1
**Iconographic Falls Efficacy Scale—mean (SD)**	31 (6.50)	--	--	26.07 (6.34)	−4.93 (−6.94, −2.93)	<0.01 *
**Quality of Life-AD—mean (SD)**						
Participant-rated	32.2 (6.75)	--	--	33.8 (6.60)	1.60 (−0.71, 3.91)	0.16
Carer-rated	28.8 (5.40)			30.93 (5.42)	2.13 (−1.02, 5.28)	0.17
**Sleep quality rating—mean (SD)**	3.07 (1.22)	--	--	2.6 (1.24)	−0.58(−1.35, 0.18)	0.14
**Healthcare use in the past 3 months—*n* (%)**						
Admitted to emergency department	0	--	--	0	N/A	
Admitted to hospital	2 (13.33)			0	0	
Consulted general practitioner	14 (93.33)			13 (86.67)	−0.77 (−2.31, 0.77)	0.33
Consulted other health professional(s)	5 (33.33)			8 (53.33)	0.83 (−0.06, 1.72)	0.07
**Falls in the past month—*n* (%)**						
0	11 (73.33)	12 (80.00)	10 (66.67)	11 (73.33)	0.13 (−0.34, 0.60)	0.55
1	1 (6.67)	2 (13.33)	3 (20.00)	1 (6.67)		
>1	3 (20.00)	1 (6.67)	2 (13.33)	3 (20.00)		

^a^ Linear regression analysis for continuous data, ordinal logistic regression for ranked data. All comparisons were made between data at baseline (T_0_) and Week 12 (T_3_), with independent variables (time-point) coded as 0 for T_0_ and 1 for T_3_. * Statistical significance at *p* < 0.05

## Data Availability

Data are available from the corresponding author upon reasonable request.

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
