# Peer review of "A Scalable Program for Improving Physical Activity in Older People with Dementia Including Culturally and Linguistically Diverse (CALD) Groups Who Receive Home Support: A Feasibility Study"

_ijerph, 2023, doi:10.3390/ijerph20043662_

Round 1

Reviewer 1 Report

The manuscript report pilot-tested a co-designed physical exercise program for 12 weeks for feasibility, safety, adherence, and the potential for benefits in terms of physical activity, physical function, healthcare utilization, and falls for people with dementia or cognitive impairment. The manuscript is clear, relevant, and significant for the field and illustrates the Australian home care system. 

The manuscript contains some writing errors; a minor revision to the writing style is suggested; some errors in abstract. 

In the section 2.1 Design a suggestion : One-group pretest-posttest design

Some subscript points in Table 1, are the information on the instruments in the section "Materials and Methods." It is confusing to have that description after the table with a subscript annotation. I propose removing this point and incorporating it into the section "Materials and Methods" with the descriptions of the instruments mentioned.

The results reinforce the importance and benefits of a structured and supported physical exercise program. 

Reviewer 2 Report

In this manuscript the authors physical exercise program for feasibility. In its current form the manuscript cannot be accepted as longer are needed in the methodology and conclusions sections.

Results and discussions were well presented and discussed.

In one of the methodologies sections, the authors must include all the professionals involved in this program. This information should include, among other things, professional training, qualification (specialist, master, etc.) and length of experience with the service offered. This information should also support the discussion of the manuscript, as the qualifications and experiences of professionals can positively impact the quality of the service offered and also the recovery of patients.

In the abstract, the authors reported "each patient's dementia/cognitive impairment". In section 2.2 the authors must describe the problems/diseases presented by the patients.

Still in the methodology, the authors must include a methodological figure including the groups of patients, type of study and any other information capable of informing in a summarized way what was done in this manuscript.

The manuscript conclusions are too bad. In these conclusions, the authors should be able to inform the main results obtained by the approaches used, in addition to informing how the results of this manuscript can serve for the future treatment of patients with similar problems.

Reviewer 3 Report

Dear Authors,

thank you for the opportunity to review this manuscript, which was very interesting and i think it will be a valuable contribution to the literature. I I was especially intrigued with your approach to manage physical therapt and exercise for older people with dementia. However i suggest to better clarify the importance and effects of physical activity in this kind of patient in the introduction section. Moreover, figures of exercises should be improved.

Reviewer 4 Report

Congratulations to the authors for the article presented.  It seems to me a very interesting and necessary work.

I think it is very well worked, although I consider that the design should be justified a little more, although it is clear.  In turn, the conclusion should be developed more, providing the authors' reflections.

Best regards

Round 2

Reviewer 2 Report

After modifications, the manuscript can be accepted for publication.